# RANGE-LIMITED AUGMENTATION FOR FEW-SHOT LEARNING IN TABULAR DATA

## ABSTRACT

Few-shot learning is essential in many applications, particularly in tabular domains where the high cost of labeling often limits the availability of annotated data. To address this challenge, we propose *range-limited augmentation* for contrastive learning in tabular domains. Our augmentation method shuffles or samples values within predefined feature-specific ranges, preserving semantic consistency during contrastive learning to enhance few-shot classification performance. To evaluate the effectiveness of our approach, we introduce FESTA (**Fe**w-**S**hot **Ta**bular classification benchmark), a benchmark consisting of 42 tabular datasets and 31 algorithms. On this benchmark, contrastive learning with our augmentation method effectively preserves task-relevant information and significantly outperforms existing approaches, including supervised, unsupervised, self-supervised, semi-supervised, and foundation models. In particular, our method achieves an average rank of 2.3 out of 31 algorithms in the 1-shot learning scenario, demonstrating its robustness and effectiveness when labeled data is highly limited. The benchmark code is available in the supplementary material.

## 1 INTRODUCTION

In many machine learning applications, obtaining labeled data presents significant challenges due to the labor-intensive nature of the labeling process (Chapelle et al., 2009). This issue is particularly relevant in tabular domains, where acquiring labeled data is often expensive and requires expert knowledge, despite the availability of abundant unlabeled data (Yoon et al., 2020; Nam et al., 2023b;a; Hegselmann et al., 2023; Han et al., 2024). For instance, during the early stages of the COVID-19 pandemic, early detection efforts were hindered by the limited availability of labeled data, such as confirmed cases, despite the abundance of related but unlabeled data (Zhou et al., 2020). This scarcity underscores the need for few-shot learning techniques that can maximize performance with minimal labeled data.

Given the scarcity of labeled data in tabular domains, contrastive learning has emerged as an effective strategy to leverage abundant unlabeled data (Bahri et al., 2021; Ucar et al., 2021; Wang & Sun, 2022; Somepalli et al., 2021). In this approach, we first learn the representations by optimizing contrastive loss with unlabeled data, then leverage the limited labeled data to train a simple prediction head by optimizing the supervised loss on these learned representations. The performance of contrastive learning significantly depends on the choice of data augmentations because it directly controls the information captured by the representations (Chen et al., 2020a; Tian et al., 2020a; Grill et al., 2020; Lee et al., 2024a). For better representation learning, augmentations should retain task-relevant information while minimizing the nuisance information (Linsker, 1988; Tian et al., 2020a; Xiao et al., 2020; Purushwalkam & Gupta, 2020). In other words, augmented views should share the same task labels after augmentation, while task-irrelevant factors can be perturbed.

Defining data augmentations that preserve task-relevant information is particularly challenging in tabular data, as it is difficult to assess whether the augmentations maintain the task labels. In contrast, in domains like images, this process is relatively straightforward; for instance, flipping or resizing an image does not alter its label in object classification. However, in tabular domains, this clarity is often unavailable. For example, in a medical dataset where the task is to predict infection status, it is unclear whether masking or shuffling certain values, such as body temperature, would preserve the task label without expert knowledge. This uncertainty complicates the design of augmentations that reliably maintain semantic information in tabular data.

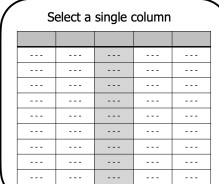 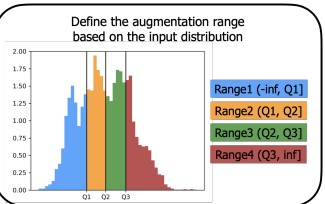 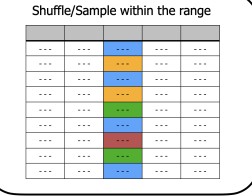

**Figure 1:** An overview of our augmentation methods, range-limited shuffling and sampling: Before training, we define the augmentation ranges for each numerical feature based on the input distribution for a given number of ranges. During training, we implement shuffling and sampling within the predefined ranges to generate augmented views. This procedure is applied to all numerical features.

Recent works suggest that grouping nearby samples in the data distribution can significantly improve downstream task performance in tabular domains. Lee et al. (2024b) demonstrated that pretraining on unlabeled datasets to predict feature quantization bins can largely improve downstream task performance. Similarly, Wu et al. (2023) proposed using randomized quantization as an augmentation strategy in contrastive learning, showing that withholding information within quantization bins enhances performance across diverse data domains. These findings imply that samples close in the data distribution can be treated as having the same values to improve tabular representation learning, possibly due to shared semantics within the same group. Building on this, we hypothesize that restricting augmentations to specific ranges based on distributional proximity (*i.e.*, proximity within feature distributions in the training data) will help preserve semantic consistency in tabular data.

Building on our hypothesis, we propose *range-limited augmentation* methods within a contrastive learning framework to enhance few-shot classification in tabular data. As illustrated in Figure 1, the main idea is straightforward: *shuffle or sample values within predefined ranges for each feature*. By limiting these ranges, our method aims to maintain semantic consistency between augmented views and original samples, providing more reliable positive pairs for contrastive learning. This approach helps reduce the risk of false positives and enhances the model's ability to learn meaningful invariances. To address the unique characteristics of tabular data, we apply feature-wise transformations to adjust ranges based on the distribution of each feature to account for different feature scales. In addition, we conduct quantitative analyses to validate our hypothesis that nearby samples share task labels, confirming that our range-limited augmentation preserves task-relevant information more effectively than existing augmentation methods.

To validate the generalizability of our method, we introduce FESTA (**Fe**w-**S**hot **Ta**bular classification benchmark), a comprehensive benchmark that evaluates 31 algorithms across 42 public tabular datasets. FESTA assesses scenarios with only a few number of labeled samples and a large pool of unlabeled data. The benchmark covers models from various learning paradigms, including supervised, unsupervised, self-supervised, and semi-supervised, and foundation models. To the best of our knowledge, FESTA is the first and largest benchmark dedicated to few-shot learning in tabular domains, providing a thorough evaluation of algorithmic performance. Our experiments on the FESTA benchmark demonstrate that our approach significantly improves few-shot classification performance over existing tabular learning methods, achieving an average rank of 2.3 out of 31 algorithms using only 1-shot labeled data.

In summary, the contributions of this paper are as follows: (1) We propose range-limited augmentation, a simple yet effective tabular augmentation strategy for contrastive learning. (2) We introduce FESTA, a comprehensive benchmark for few-shot learning in tabular domains, evaluating 31 algorithms across 42 public datasets. The benchmark code is available in the supplementary material. (3) Our method consistently and significantly improves few-shot classification performance across various numbers of labeled samples and datasets.

## 2 RELATED WORK

**Learning with few labeled samples:** Prior works on learning with limited labeled data leverage unlabeled samples through two main approaches: semi-supervised (Lee et al., 2013; Kim et al., 2020; Assran et al., 2021; Pham et al., 2021) and self-supervised (Chen et al., 2020b;a; 2021a; Yue et al., 2021) approaches. Semi-supervised learning often employs pseudo-labeling, where model predictions on unlabeled data are used as labels during training (Lee et al., 2013). To improve pseudo-labeling quality, recent advancements have introduced momentum networks (Laine &

Aila, 2016; Tarvainen & Valpola, 2017; Pham et al., 2021) and consistency regularization through data augmentations (Berthelot et al., 2019b;a; Sohn et al., 2020; Xie et al., 2020). In contrast, self-supervised learning focuses on learning representations using domain-specific inductive biases, such as spatial relationships in images and temporal relationships in time-series data, followed by fine-tuning on the few available labeled samples (Tian et al., 2020b; Perez et al., 2021). Notably, self-supervised methods have demonstrated strong performance in transductive settings, often outperforming conventional few-shot learning techniques (Chen et al., 2021b; Nam et al., 2023b). Both semi-supervised and self-supervised approaches rely heavily on effective data augmentations. Although some augmentations have been developed specifically for tabular data, their effectiveness in few-shot learning settings remains underexplored. To address this, we introduce range-limited augmentation tailored for contrastive learning to enhance few-shot classification in tabular data.

**Learning with unlabeled samples in tabular domains:** Recent efforts have explored leveraging unlabeled data to enhance model performance in tabular domains when labeled samples are limited. For instance, Yoon et al. (2020) introduced a self-supervised and semi-supervised framework using a novel augmentation that masks feature values to train an encoder. Building on this, Bahri et al. (2021) developed a contrastive learning approach, randomizing feature values based on empirical marginal distributions, while Ucar et al. (2021) proposed multi-view representation learning by splitting features into subsets. In another direction, Lee et al. (2024b) suggested a pretext task that predicts bin indices to capture dataset irregularities, with random shuffling improving downstream performance. Beyond augmentations, Nam et al. (2023b) explored unsupervised meta-learning, using self-supervised tasks from unlabeled data for few-shot classification. Other recent works (Nam et al., 2023a; Hegselmann et al., 2023; Han et al., 2024) leveraged large language models (LLMs) to utilize in-context learning on unlabeled datasets. In our study, we focus on methods that operate without relying on auxiliary information, such as column descriptions.

**Data augmentation in tabular contrastive learning:** Data augmentation is essential in contrastive learning for generating positive views that enable the model to learn meaningful invariances. However, unlike image or time-series data with clear spatial or temporal structures, tabular data lacks such inductive biases, complicating the design of augmentations that both preserve task-relevant information and introduce useful perturbations. Current augmentation techniques for contrastive learning in tabular data can be grouped as follows:

- Masking (Yoon et al., 2020; Huang et al., 2020): Randomly mask feature values with a constant.
- Sampling (Bahri et al., 2021): Randomly replace feature values based on their empirical marginal distributions.
- Shuffling (Huang et al., 2020; Lee et al., 2024b): Shuffle feature values within each feature column.
- Noise (Nam et al., 2023b): Inject small random noise into selected feature values.
- Subset (Ucar et al., 2021; Wang & Sun, 2022): Divide the input features into multiple subsets.
- CutMix (Somepalli et al., 2021): Combine two samples using a binary mask applied to feature values.
- MixUp (Somepalli et al., 2021): Linearly interpolate between a sample and a randomly selected sample from the same batch in the embedding space.
- Random quantization (Wu et al., 2023): Quantize each feature channel into uniform or non-uniform bins and replace feature values with random constants within these bins.

A detailed description of each augmentation is provided in Supplementary A.3. In this study, we introduce two new augmentation methods aimed at better preserving semantic information to improve few-shot classification performance in tabular data.

## 3 FeSTa: Few-shot Tabular Classification Benchmark

In this section, we introduce FeSTa (**Fe**w-**S**hot **Ta**bular classification benchmark), a comprehensive benchmark designed to evaluate the performance of few-shot classification algorithms in tabular domains. The benchmark encompasses 42 public datasets and 31 algorithms for a thorough evaluation of our proposed method, as well as existing approaches. FeSTa spans multiple learning paradigms, including supervised, unsupervised, self-supervised, and semi-supervised learning, and foundation models, as well as both traditional machine learning and deep learning approaches. By providing

a diverse range of datasets and algorithms, the benchmark allows for a thorough and systematic evaluation of few-shot learning performance in tabular domains.

### 3.1 PROBLEM SETUP: FEW-SHOT SEMI-SUPERVISED CLASSIFICATION

We first describe the problem setup of our interest, the few-shot learning in tabular domains. Formally, our goal is to train a neural network classifier $f_\theta : \mathcal{X} \to \mathcal{Y}$ parameterized by $\theta$ where $\mathcal{X} \subseteq \mathbb{R}^d$ and $\mathcal{Y} = \{0, 1\}^C$ are input and label spaces with $C$ classes, respectively. We assume that we have a labeled dataset $\mathcal{D}_l = \{\mathrm{x}_{l,i}, \mathrm{y}_{l,i}\}_{i=1}^{N_l} \subseteq \mathcal{X} \times \mathcal{Y}$ and an unlabeled dataset $\mathcal{D}_u = \{\mathrm{x}_{u,i}\}_{i=1}^{N_u} \subseteq \mathcal{X}$ for training the classifier $f_\theta$. Following the convention of the few-shot learning, we set $N_l = C \times S$ where $S$ represents the number of labeled samples per class (shots). All data points are drawn from a distribution $p(\mathrm{x}, \mathrm{y})$ in an *i.i.d.* manner. We do not allow the use of auxiliary information like column descriptions or additional domain knowledge.

### 3.2 FeSTa: FEW-SHOT TABULAR CLASSIFICATION BENCHMARK

**Datasets:** We collected 42 public datasets from the OpenML Python library (Vanschoren et al., 2014), as a subset of the largest tabular learning benchmarks (McElfresh et al., 2023; Salinas & Erickson, 2023). The selection criteria were: (1) datasets contain at least one numerical feature, and (2) each class includes more than $S$ samples. The benchmark includes 26 binary and 16 multiclass classification datasets, with sizes ranging from 180 to over 250,000 samples and feature dimensions from 4 to 216. Following Nam et al. (2023b), we split each dataset into an 80% training set and 20% test set, with 10% of the unlabeled training data used for validation when necessary. A quantile transformation is applied to all numerical features for normalization. Categorical features were determined as those with fewer than 20 unique values (Lee et al., 2024b). No additional labeled data is used for training or hyperparameter optimization, ensuring the constraints of the few-shot learning setup. A complete list of datasets is provided in Supplementary A.1.

**Baselines:** We evaluate a variety of baseline algorithms spanning multiple learning paradigms to ensure a comprehensive assessment of few-shot learning in tabular data. These include:

- Supervised algorithms: Logistic regression (LR), $k$-nearest neighbors (kNN), XGBoost (Chen & Guestrin, 2016), CatBoost (Prokhorenkova et al., 2018), LightGBM (Ke et al., 2017), MLP
- Self-supervised algorithms: Reconstruction-based auto-encoder, Binning (Lee et al., 2024b), SubTab (CL+Subset, (Ucar et al., 2021)), VIME (Yoon et al., 2020), Contrastive learning with four augmentation methods (CL+Masking/Shuffling/Noise/RQ), SCARF(CL+Sampling, (Bahri et al., 2021)), SAINT (CL+CutMix+MixUp, (Somepalli et al., 2021))
- Semi-supervised algorithms: VIME (Yoon et al., 2020), Pseudo-label (Lee et al., 2013) with six augmentation methods (PL+Masking/Shuffling/Noise/RQ/Sampling/CutMix), Auto-Encoder, ICT (Verma et al., 2022), Mean Teacher (Tarvainen & Valpola, 2017)
- Unsupervised meta-learning algorithm: STUNT (Nam et al., 2023b)
- Foundation models: TabPFN (Hollmann et al., 2022), HyperFast (Bonet et al., 2024)

In addition to these baselines, our benchmark includes two self-supervised learning methods incorporating our new data augmentation techniques. Due to the limited number of labeled samples for training and validation, we directly apply the best setups for each model as reported in the original papers, without tuning hyperparameters. For self-supervised learning algorithms, we primarily use logistic regression as the evaluation protocol in the manuscript, as it shows the best performance across datasets. Alternative evaluation methods, including $k$-nearest neighbors, linear evaluation, and fine-tuning, are also available in the benchmark, with full details provided in Supplementary C.1. Following Nam et al. (2023b), a 2-layer MLP is used as the classifier $f_\theta$ for most deep learning algorithms if there is no specific architecture is provided in the original papers. Detailed descriptions and configurations for each algorithm are provided in Supplementary A.2.

**Evaluation:** For each dataset and algorithm, we use 50 different data splits to evaluate performance. We evaluate accuracy for $S \in \{1, 5\}$ across all datasets, but AUROC and log loss results are also available in the benchmark.

## 4 RANGE-LIMITED AUGMENTATION FOR FEW-SHOT TABULAR LEARNING

In this study, we leverage contrastive learning framework to make effective use of unlabeled data for few-shot learning. Specifically, we train an encoder on unlabeled data to learn representations

that capture useful invariances through data augmentations, followed by training a simple prediction head (*e.g.*, logistic regression) on the limited labeled data. The performance of contrastive learning heavily depends on data augmentations, as they control the information captured by the representations (Chen et al., 2020a; Tian et al., 2020a; Grill et al., 2020; Lee et al., 2024a). Effective augmentations should retain task-relevant information while reducing nuisance factors (Linsker, 1988; Tian et al., 2020a; Xiao et al., 2020; Purushwalkam & Gupta, 2020), ensuring augmented views share the same task labels.

However, tabular data lacks clear inductive biases, making it challenging to design augmentations that preserve task-relevant information. For example, masking or shuffling values can disrupt semantic relationships and lead to false positive pairs, hindering contrastive learning from capturing meaningful invariances. Recently, several studies found that grouping nearby samples based on their proximity in the data distribution can improve the downstream task performance. Lee et al. (2024b) found that pretraining to predict feature quantization bins, rather than raw values, improves downstream task performance, while Wu et al. (2023) used randomized quantization to make feature values constant within the same bins as an augmentation strategy. These findings suggest that samples close in the data distribution benefit from being treated similarly during training, potentially due to shared semantics within each group. Building on this, we hypothesize that restricting augmentations to predefined ranges based on distributional proximity can better preserve task-relevant information, thereby enhancing few-shot classification.

**Range-limited augmentation:** The main idea is straightforward: we shuffle or sample values within predefined ranges for each feature. As shown in Figure 1, each feature is divided into $b$ ranges, ensuring that each range contains an equal number of observations (Wu et al., 2023; Lee et al., 2024b). For a given input sample $\mathbf{x}$, we generate an augmented view $\mathbf{x}'$ based on the augmentation ranges $\mathbf{B}_j = \{B_{j1}, B_{j2}, \ldots, B_{jb}\}$ for each feature $j \in [1, d]$, where each range $B_{jk} = (\beta_{jk}^{\min}, \beta_{jk}^{\max}]$ is defined by its boundaries.

- **Range-limited shuffling:** We shuffle the values within the same range. For the $i$-th sample of the $j$-feature, $x_{i,j} \in B_{jk}$, the augmented value is sampled from the set of values within the same range: $x'_{i,j} \sim \{v | v \in x_{\cdot,j} \text{ and } v \in B_{jk}\}$.

- **Range-limited sampling:** We sample values from a uniform distribution bounded by the range limits. For the $i$-th sample of the $j$-feature, $x_{i,j} \in B_{jk}$, the augmented value is drawn as $x'_{i,j} \sim \mathcal{U}(\beta_{jk}^{\min}, \beta_{jk}^{\max})$.

The range-limited augmentation is applied to randomly selected cells in each sample, controlled by a hyperparameter called the selection ratio, $p \in [0, 1]$. Specifically, a Bernoulli distribution with probability $p$ generates a masking vector $\mathbf{m} \in \{0, 1\}^d$ of the same size as $\mathbf{x}$. The final transformed sample is generated as $\mathbf{x}_{\text{aug.}} = \mathbf{m} \odot \mathbf{x}' + (1 - \mathbf{m}) \odot \mathbf{x}$.

**Overall framework:** Our framework follows a conventional self-supervised learning pipeline. We first train an encoder by optimizing a SimCLR-like contrastive loss (Chen et al., 2020a) on unlabeled data, using range-limited augmentation to generate positive pairs. After pretraining, a logistic regression prediction head is trained on top of the frozen encoder using the few labeled samples available. Consistent with prior works (Yoon et al., 2020; Bahri et al., 2021; Nam et al., 2023b), the encoder is trained for 1000 epochs with early stopping, and we set $p = 0.3$ throughout our study. For both range-limited shuffling and sampling, we fix the number of ranges $b$ as 4 throughout our study. This choice maintains a balance between preserving task-relevant information and computational efficiency. (See Section 6.2 for a detailed empirical analysis.)

### 4.1 ANALYSIS OF TASK-RELEVANT INFORMATION PRESERVED BY AUGMENTATION METHODS

Evaluating how well augmentation preserves task-relevant information is challenging in tabular domains, as the labeling process often requires additional steps or expert knowledge. To address this challenge, we use a pretrained neural classifier $f_\theta$ with near-perfect test accuracy as a proxy for the ground-truth labeling process. This classifier is trained on the original training samples (without augmentation) and evaluated on transformed test samples. It is considered as a reliable proxy when it achieves over 95% accuracy on the original test samples, which we observe in a subset of eight datasets within our benchmark. Using this proxy, we test our hypothesis that range-limited augmentation preserves task-relevant information more effectively than six previous augmentation methods:

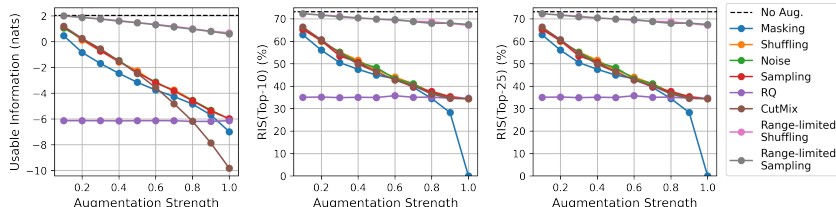

**Figure 2:** Comparison of usable information (Left) and representation invariance score (Middle, Right) across different augmentation methods and augmentation strengths: Most augmentations tend to reduce both metrics as augmentation strength increases, indicating a loss of task-relevant information. In contrast, range-limited augmentations consistently preserve high levels of both metrics across all augmentation strengths, outperforming other methods and demonstrating their efficacy in retaining task-relevant information.

masking, shuffling, sampling, noise, CutMix, and RQ, across varying augmentation strengths. Here, augmentation strength refers to how many cells are affected by the augmentation function, such as the selection ratio $p$ for masking, shuffling, sampling, noise, CutMix, and our method, or the quantization scale for RQ.

To evaluate which augmentation is better to preserve task-relevant information than others, we measure two metrics on the representation $Z$ from the penultimate layer of $f_\theta$. The classifier $f_\theta$, trained without augmentations, serves as a proxy to assess how much task-relevant information is retained in $Z$ for transformed test datasets.

- Usable information (Kleinman et al., 2020): It quantifies the relevant information in $Z$, the representation of augmented inputs, for predicting the target label $Y$. It is defined as $I(Z; Y) = H(Y) - L_{CE}(p(y|z), q(y|z))$, where $H(Y)$ is the entropy of $Y$, and $L_{CE}$ is the cross-entropy loss between the predicted distribution $q(y|z)$ and the true distribution $p(y|z)$. A higher value indicates that the representation $Z$ retains more relevant information to predict target label $Y$, thereby the augmentation preserves task-relevant information well.

- Representation Invariance Score (RIS) (Goodfellow et al., 2009; Purushwalkam & Gupta, 2020): RIS measures the consistency of $Z$ under augmentations based on $Y$. It is calculated as the average consistency of the activation patterns across the top-$K$ units of $Z$ for each class. A higher RIS suggests that an augmentation maintains consistent activation patterns in the representations $Z$ for the same task label $Y$, thereby preserving task-relevant information more effectively than augmentations that disrupt these patterns.

Figure 2 shows the effect of augmentation strength on usable information and RIS across various augmentation methods. Most augmentations exhibit a clear decline in both metrics as augmentation strength increases, suggesting a disruption in retaining task-relevant information, while RQ maintains consistently low levels for both metrics, potentially due to a sensitivity to even minor transformations. In contrast, range-limited augmentations – both shuffling and sampling – maintain robust performance across all strengths. They consistently preserve high levels of usable information and RIS, indicating that task-relevant information is well-retained even at higher augmentation levels. These results demonstrate the efficacy of our range-limited methods in preserving task-relevant information while maintaining consistent representations across varying augmentation strengths.

## 5 EXPERIMENTS

In this section, we conduct extensive experiments to evaluate how our range-limited augmentation methods improve few-shot classification performance on the FESTA benchmark. Performance is evaluated on 50 random splits per dataset, with results reported as averages and standard deviations. (Full results are available in the FESTA benchmark and are also provided in the zip file included in the supplementary materials during the review process.) All experiments are performed on a single NVIDIA GeForce RTX 3090. Our results show that our methods consistently and significantly outperform existing approaches, including supervised, unsupervised, self-supervised, semi-supervised, and foundation models. These findings underscore the importance of preserving task-relevant information during contrastive learning to enhance few-shot classification performance.

For few-shot classification, we assess performance under 1-shot and 5-shot scenarios, where one or five labeled samples per class are available. As summarized in Table 1, we found that unsupervised,

**Table 1:** Experimental results on the FESTA benchmark: We evaluate each algorithm's performance on 50 different data splits per dataset and report the average accuracy and standard deviation. The average rank is calculated based on average accuracy across datasets. The "Wins" column indicates how often each algorithm achieves the highest average accuracy for a dataset, with ties counted. The best-performing algorithm for each number of labeled samples (1-shot and 5-shot) and metric is highlighted in **bold**. Since TabPFN is incompatible with large datasets, we also compare results on a subset of smaller datasets, indicated by †. Despite modifying only the augmentation module, our method significantly outperforms other baselines.

| Shots | | 1 | | | 5 | | |
|---|---|---|---|---|---|---|---|
| Model | | Accuracy (%) | Rank | Wins | Accuracy (%) | Rank | Wins |
| Supervised | LR | 48.569±15.525 | 9.262±4.949 | 1 | 57.567±17.385 | 9.881±6.978 | 0 |
| | kNN | 49.116±14.499 | 9.095±4.482 | 0 | 54.210±16.135 | 17.167±6.522 | 0 |
| | XGBoost | 41.328±23.757 | 18.167±11.144 | 9 | 57.199±15.312 | 15.119±7.409 | 1 |
| | CatBoost | 48.345±16.045 | 10.095±7.091 | 6 | 59.522±18.330 | 10.429±7.847 | 4 |
| | LightGBM | 41.689±23.240 | 18.119±11.123 | 9 | 50.267±19.893 | 20.500±10.639 | 6 |
| | MLP | 48.269±15.224 | 9.643±4.405 | 0 | 57.996±17.592 | 10.333±5.937 | 1 |
| Semi-Supervised | VIME | 41.505±13.492 | 21.238±8.310 | 1 | 50.944±15.019 | 20.333±6.362 | 0 |
| | Auto-Encoder | 47.353±15.744 | 12.119±7.249 | 1 | 57.020±18.396 | 12.048±6.953 | 1 |
| | ICT | 44.576±13.655 | 16.690±8.236 | 0 | 50.926±18.658 | 18.881±7.164 | 0 |
| | MeanTeacher | 45.527±13.861 | 15.952±6.604 | 1 | 54.422±16.269 | 17.500±6.302 | 1 |
| | PL+Masking | 43.987±13.905 | 18.571±6.232 | 0 | 55.180±17.561 | 14.690±6.798 | 0 |
| | PL+Sampling | 43.690±13.977 | 20.500±6.452 | 0 | 53.515±17.484 | 17.571±6.275 | 0 |
| | PL+Shuffling | 43.610±13.634 | 20.405±6.073 | 0 | 52.924±17.565 | 18.310±7.220 | 0 |
| | PL+Noise | 43.417±13.824 | 20.714±5.148 | 0 | 53.006±17.369 | 17.738±7.408 | 0 |
| | PL+RQ | 44.698±14.577 | 17.524±6.812 | 0 | 54.084±17.878 | 16.143±6.906 | 0 |
| | PL+CutMix | 43.212±13.488 | 21.786±6.437 | 0 | 53.085±17.063 | 18.524±6.751 | 0 |
| Unsup. Meta | STUNT | 46.955±15.471 | 13.381±7.119 | 1 | 53.412±16.903 | 15.095±8.720 | 2 |
| Foundation | HyperFast | 47.798±13.615 | 14.732±6.565 | 0 | 59.772±18.736 | 8.310±6.119 | 3 |
| Self-supervised | Reconstruction | 33.414±16.978 | 27.810±2.957 | 0 | 32.816±17.381 | 28.976±2.136 | 0 |
| | Binning | 34.564±17.248 | 27.071±4.474 | 0 | 34.114±16.994 | 28.238±4.029 | 0 |
| | VIME | 35.999±17.520 | 26.476±3.833 | 0 | 36.428±18.166 | 27.405±3.328 | 0 |
| | SubTab (CL+Subset) | 36.264±17.614 | 26.190±3.921 | 0 | 36.680±18.005 | 28.262±2.548 | 0 |
| | SCARF (CL+Sampling) | 48.830±14.716 | 11.024±5.470 | 0 | 59.170±16.073 | 12.262±5.579 | 1 |
| | SAINT (CL+CutMix+MixUp) | 45.191±18.857 | 16.143±7.863 | 1 | 50.768±20.715 | 18.571±8.279 | 2 |
| | CL+Masking | 48.114±14.885 | 11.714±4.815 | 0 | 56.787±17.365 | 14.333±5.011 | 0 |
| | CL+Shuffling | 49.091±14.899 | 10.524±5.379 | 1 | 59.373±16.233 | 11.238±6.084 | 0 |
| | CL+Noise | 49.076±14.881 | 10.738±5.747 | 1 | 59.394±16.263 | 11.167±5.725 | 1 |
| | CL+RQ | 47.153±16.012 | 12.929±5.242 | 0 | 55.882±18.437 | 13.381±5.780 | 0 |
| | CL+Range-limited Shuffling | **51.972±15.243** | **2.310±1.405** | **16** | **61.921±16.641** | **3.857±3.302** | **16** |
| | CL+Range-limited Sampling | 50.640±14.759 | 4.857±2.455 | 2 | 60.647±16.315 | 7.738±4.580 | 0 |
| Self-supervised | CL+Range-limited Shuffling† | 52.670±15.038 | 2.303±1.447 | 13 | 60.827±16.523 | 3.636±3.525 | 14 |
| | CL+Range-limited Sampling† | 51.284±14.534 | 4.848±2.563 | 2 | 59.610±16.214 | 7.727±4.824 | 0 |
| Foundation | TabPFN† | 48.216±13.631 | 14.727±6.256 | 1 | 60.421±15.921 | 6.394±4.815 | 3 |

self-supervised, and semi-supervised models do not consistently outperform supervised baselines in both setups, despite access to large amounts of unlabeled data. This suggests that current few-shot learning techniques cannot effectively leverage unlabeled data to capture task-relevant dependencies.

However, substituting the augmentation module in a self-supervised framework with our range-limited augmentation yields significant performance improvements in both 1-shot and 5-shot scenarios. Specifically, our method achieves an average rank of 2.3 out of 31 algorithms in the 1-shot setup, highlighting the critical role of preserving task-relevant information during contrastive learning for enhancing few-shot classification across various datasets. While our approach incurs a slight increase in training time due to the overhead of range-limited augmentations, it achieves superior performance with only a minimal additional cost compared to more complex architectures like transformers. Additional details on training time can be found in Supplementary C.2.

While most methods use MLP-based architectures, transformer-based models like SAINT and TabPFN are included for comparison. Interestingly, no consistent advantage of transformer architectures over MLPs is observed in few-shot settings, suggesting that model architecture alone does not indicate better performance when labels are scarce.

Surprisingly, contrastive learning with range-limited augmentation outperforms foundation models such as TabPFN and HyperFast, both trained on large-scale tabular datasets, while our approach is trained on a single dataset. Since TabPFN has constraints to use, including sample size, feature dimension, and number of classes, we compared the effectiveness of our method with TabPFN on a subset of the FESTA benchmark consisting of 33 datasets, in the bottom three lines of Table 1. Importantly, both foundation models require a small number of labeled samples for inference, such as to construct the attention maps and determine model weights. Our approach demonstrates an average accuracy improvement of 4.19% over TabPFN and 3.95% over HyperFast in the 1-shot

setting, even though the foundation models leverage large-scale datasets and complex architectures, whereas our method employs a simple 2-layer MLP trained on a single dataset. This underscores the importance of augmentations that preserve task-relevant information, enabling effective learning of latent relationships in tabular data without relying solely on large-scale training.

Among our methods, range-limited shuffling consistently outperforms range-limited sampling. A similar trend is observed when comparing the performance of CL+Shuffling with SCARF (CL+Sampling). These results suggest that generating augmented views with values already present in the dataset can be more beneficial than sampling new values for tabular representation learning. Nonetheless, range-limited sampling proves to be highly competitive, outperforming all other few-shot learning techniques in the benchmark. This result highlights the superiority of range-limited augmentations in enhancing few-shot classification.

## 6 DISCUSSION

We have observed that our method significantly improves few-shot classification performance across a wide range of datasets. In this section, we present additional experiments, with evaluations conducted on 10 random splits per dataset, to further investigate the advantages of our approach.

### 6.1 INCREASING THE NUMBER OF LABELED SAMPLES

Beyond the original evaluation with the number of labeled samples $N_l = S \times C$ and $S \in \{1, 5\}$, we examine the performance of top-performing algorithms from Table 1 as the number of shots increases. As summarized in Figure 3, all algorithms perform better with an increasing number of labeled samples. Interestingly, the 5-shot performance of our method is comparable to that of other algorithms with 10 or more shots, and its superiority persists even as the number of shots increases. These results highlight the ability of our method to effectively capture task-relevant information, demonstrating superior downstream task performance even as the number of shots increases.

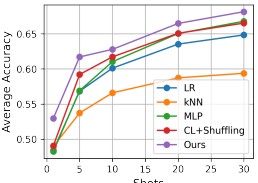

**Figure 3:** Experimental results increasing the number of labeled samples: Our method (CL+Range-limited shuffling) consistently achieves superior performance with an increased number of shots.

### 6.2 ABLATION STUDY: SELECTION RATIO AND THE NUMBER OF RANGES

In this study, we fixed the selection ratio $p = 0.3$ and the number of ranges $b = 4$ for our augmentation methods, using optimal hyperparameters for other augmentation techniques as suggested in their respective papers. However, as demonstrated in Section 4.1, augmentation strength affects the amount of shared task-relevant information between views. This strength is controlled by the hyperparameter $p$ in our methods as well as other augmentations like masking, shuffling, sampling, and noise, where $p$ defines the proportion of cells augmented.

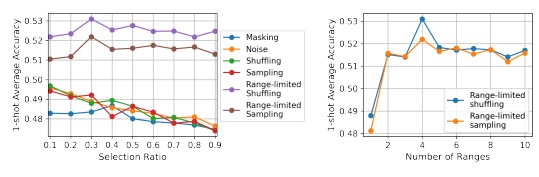

**(a)** Selection ratio $p$      **(b)** Number of ranges $b$

**Figure 4:** Ablation results showing the effect of (a) varying the selection ratio $p$ across different augmentation strategies and (b) varying the number of ranges $b$ for range-limited augmentations.

To evaluate the effect of $p$, we conducted experiments exclusively on augmentation methods that use $p$ as a hyperparameter, as summarized in Figure 4a. Our results show that our range-limited augmentation consistently achieves superior performance across different values of $p$, indicating its robustness to varying augmentation strengths. In contrast, other methods degrade in performance as $p$ increases, consistent with the findings in Section 4.1. In particular, the worst performance of CL+Range-limited augmentation even outperforms the best performance of all other CL+Aug methods, underscoring the robustness of our approach regardless of the choice of $p$. These observations highlight the critical role of preserving task-relevant information for effective contrastive learning.

In addition, we examined how the number of ranges $b$ affects the performance of range-limited augmentations when $p = 0.3$. While preserving task-relevant information is crucial, generating diverse views (Wang & Isola, 2020) and reducing task-irrelevant information (Xiao et al., 2020) also

**Table 2:** Experimental results for few-shot regression tasks: Performance is evaluated using the average root-mean squared error (RMSE) across three datasets. Our method consistently outperforms baseline algorithms, demonstrating better generalization with limited labeled samples in regression tasks.

| OpenML ID | 1-shot | | | | 5-shot | | | |
|---|---|---|---|---|---|---|---|---|
| | XGBoost | MLP | Ours (Shuffling) | Ours (Sampling) | XGBoost | MLP | Ours (Shuffling) | Ours (Sampling) |
| 194 | 1.839 | 1.541 | 1.537 | **1.530** | 1.621 | 1.601 | **1.539** | 1.541 |
| 44133 | 66.534 | 51.107 | **51.077** | 51.095 | 54.777 | 53.022 | 51.066 | **51.061** |
| 566 | 771.539 | 765.045 | 762.104 | **762.100** | 826.758 | 807.124 | **762.098** | 762.101 |

can contribute to better representations in contrastive learning. Increasing $b$ helps maintain task-relevant information by narrowing the augmentation ranges, but at the cost of reduced diversity. In Figure 4b, we empirically found that setting $b = 4$ provides an optimal balance, ensuring sufficient preservation of task-relevant information while maintaining diverse views. However, any choice of $b > 1$ outperforms the best performance of all other CL+Aug methods, reducing the need for extensive tuning of $b$ to achieve strong performance.

### 6.3 FEW-SHOT REGRESSION TASKS

While our primary focus has been on few-shot learning in classification tasks, our method also can be applied to regression tasks. As in the classification setting, we first train the encoder network without label information and subsequently adapt a prediction head using a few labeled samples, optimizing the supervised loss function, which in this case is the mean squared error (MSE). Since logistic regression is not suitable for regression, we employ a single linear layer for evaluation (*i.e.*, linear evaluation). We evaluated our method on three datasets from OpenML (Vanschoren et al., 2014) and measured performance based on the average root mean squared error (RMSE). For comparison, we examine two supervised baselines, XGBoost and MLP, which have demonstrated strong performance in classification tasks and are applicable to regression tasks. As summarized in Table 2, our method achieves superior performance by a substantial margin, demonstrating its effectiveness in regression tasks as well.

### 6.4 SEMI-SUPERVISED LEARNING

Data augmentation plays a crucial role not only in self-supervised learning but also in semi-supervised learning. In tabular semi-supervised learning (Yoon et al., 2020), a supervised loss is optimized with a few labeled samples, while an unsupervised loss is simultaneously optimized with unlabeled samples. For unlabeled samples, pseudo-labels are often generated from original samples and used as supervised targets for augmented samples. In this setting, preserving task-relevant information also can be important, as the representations from the original and augmented samples should correspond to the same target label. To investigate the effectiveness of range-limited shuffling in a semi-supervised learning context, we compare six pseudo-label-based semi-supervised learning methods

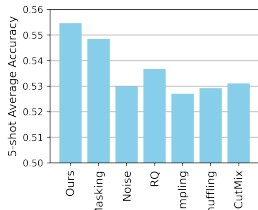

**Figure 5:** Experimental results for different augmentation methods in pseudo-label-based semi-supervised learning: Our method, range-limited shuffling, outperforms other augmentation strategies in the semi-supervised learning context.

that differ only in their choice of augmentation. The results, shown in Figure 5, present the average accuracy for the 5-shot setup. Our augmentation method outperforms all other augmentation strategies, indicating its potential advantage in semi-supervised learning.

### 7 CONCLUSION

In this study, we introduced range-limited augmentation methods tailored for few-shot learning in tabular domains. Through comprehensive evaluation on the FESTA benchmark, we demonstrated that our approach significantly outperforms existing methods. By effectively preserving task-relevant information during contrastive learning, our range-limited augmentations improve representation learning and enhance few-shot classification performance, even when labeled data is scarce. While our focus was on preserving task-relevant information, reducing nuisance factors also plays a crucial role in the design of effective data augmentations. Additionally, our work primarily addressed augmentation methods for numerical features and did not consider augmentations specifically designed for categorical features. We hope that our study can serve as a valuable starting point for future exploration in developing augmentation methods that balance both preserving meaningful information and mitigating nuisance factors.

**Ethics Statement:** This study presents range-limited augmentation techniques to enhance few-shot learning for tabular data. Our research does not involve human subjects or personally identifiable information, minimizing direct ethical concerns related to privacy or data misuse. However, as our methodologies are evaluated on open-source datasets, we have ensured that all data used is publicly available, properly cited, and compliant with OpenML licensing (CC-BY license). Our approach could be applied across a range of domains, potentially including sensitive applications such as healthcare or finance. While our methods are designed to improve generalizability and robustness, any application to such sensitive domains should consider the ethical implications, including fairness, transparency, and unintended biases in model performance. Moreover, we acknowledge that advancements in model performance can have both positive and potentially harmful applications, and we encourage the responsible use of this technology in alignment with ethical AI principles.

**Reproducibility Statement:** To ensure the reproducibility of our results, we have provided comprehensive details on the datasets, experimental setups, and baselines in the main text and supplementary materials. The full list of datasets and their descriptions is available in Supplementary A.1, while the detailed algorithm configurations, hyperparameters, and architecture settings are described in Supplementary A.2 and A.3. All experiments were conducted with a single NVIDIA GeForce RTX 3090, as specified in the paper. We also provide the code for implementing our augmentation methods and benchmarking across multiple baselines and datasets. This code and results, along with any additional instructions for reproducing the experiments, will be submitted as a zip file in the supplementary materials during the review process.

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

# SUPPLEMENTARY MATERIALS

# A BENCHMARK AND EXPERIMENTAL SETUP DETAILS

In this study, we introduce a comprehensive few-shot tabular classification benchmark, called FESTA, encompassing 42 public datasets and 31 algorithms. All datasets can be easily loaded from OpenML (Vanschoren et al., 2014) Python library (CC-BY license) with data IDs. The benchmark codes are available in the .zip file for review process, and it will be publicly available in GitHub repository. To implement the baseline algorithms, we follow the optimal setups as reported in the original papers or code repositories, ensuring the constraints of the few-shot learning setup. If there is no specific description about the choice of deep learning architectures, we use a 2-layer MLP with the layer widths as 1024, following Nam et al. (2023b). More detailed description and setups are provided as follows.

## A.1 DATASETS

Our benchmark encompasses a comprehensive collection of 42 datasets, all of which are publicly accessible through the OpenML Python library (Vanschoren et al., 2014). All datasets from OpenML are provided under the CC-BY license, which implies that the data is publicly available and has been shared with the appropriate consent and ethical considerations. OpenML ensures that datasets shared on their platform comply with their data-sharing guidelines, which include obtaining necessary consent where applicable.

We provide a detailed list with corresponding OpenML dataset IDs for quick reference as follows. Each dataset can be loaded by inserting the dataset IDs in `openml.datasets.get_dataset(DATASET_ID)`.

- 22, 54, 1063, 1067, 12, 18, 23, 59, 188, 307, 1043, 1459, 1475, 1489, 1492, 1497, 1503, 4153, 40499, 44125, 44131, 45062, 44157, 1462, 44160, 29, 37, 53, 49, 1504, 1494, 41143, 44126, 40981, 41168, 44091, 44158, 44123, 44090, 40922, 44161, 45714

The benchmark includes 26 binary and 16 multiclass classification datasets. As shown in Figure 6, data sizes ranges from 180 to over 250,000 samples and feature dimensions ranges from 4 to 216.

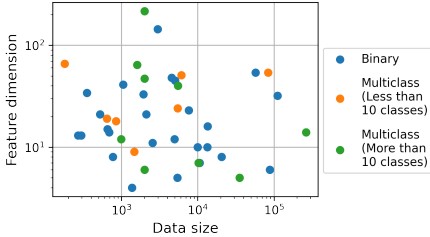

**Figure 6:** A statistical overview of FESTA benchmark: Each dot represents a dataset, with the x-axis showing data size and the y-axis representing feature dimension.

## A.2 BASELINES

We provide brief explanations of the considered baselines and the hyperparameters of the baselines. If there is no specific description for the hyperparameters in the original paper or the official code repository, we utilize the common setup of using AdamW optimizer (Loshchilov & Hutter, 2017) with learning rate $1e^{-3}$ and batch size of 100, ReLU activation, and 100 epochs with 2-layer MLP, following the setup of (Nam et al., 2023b). For all baselines, the detailed configurations are available in `config/` directory in the benchmark repository.

For our method, CL+Range-limited augmentations, we follow the default setup of 2-layer MLP with early stopping as summarized in the main text. All CL+Aug methods follow the training setup of Bahri et al. (2021), and we set $b = 4$ and $p = 0.3$ throughout our study.

### A.2.1 SUPERVISED ALGORITHMS

In supervised algorithms, we train the model using only $N_l = S \times C$ samples, where $S$ is the number of shots, and $C$ is the number of classes.

**Logistic regression:** We utilize the default settings of the scikit-learn implementation.

**$k$-nearest neighbors:** We utilize the default settings of the scikit-learn implementation. As default, we set $k$ as same as the number of shots $S$ for each task.

**XGBoost:** XGBoost (Chen & Guestrin, 2016) is an optimized distributed gradient boosting method designed to be highly efficient, flexible, and portable. We adopt the default hyperparameters provided in the XGBoost python library, with the following exceptions: `n_estimators` as 2000, `max_depth` as 10, and `eta` as 0.001, allowing for deeper trees and slower learning.

**CatBoost:** CatBoost (Prokhorenkova et al., 2018) is a fast, scalable, and high-performance gradient boosting on decision trees. We use the default hyperparameter setting in the CatBoost python library, modifying `n_estimators` to 2000, `depth` to 10, and `eta` to 0.001 to match the settings of XGBoost.

**LightGBM:** LightGBM (Ke et al., 2017) is a highly efficient gradient boosting framework designed for fast and accurate performance, using a histogram-based algorithm. We use the default hyperparameter setting in the LightGBM python library but set `n_estimators` as 2000, `max_depth` as 10, and `eta` as 0.001, consistent with XGBoost and CatBoost for fair comparison.

**MLP:** Following Nam et al. (2023b), we use a 2-layer MLP with hidden size of 1024. A dropout rate of 0.1 is applied for regularization, as recommended by Gorishniy et al. (2021). For training, we employ a cosine annealing scheduler to adjust the learning rate, as used in Lee et al. (2024b).

### A.2.2 SELF-SUPERVISED ALGORITHMS

In self-supervised algorithms, we leverage both labeled and unlabeled datasets without using label information for pre-training. Once pre-training is completed, we evaluate the learned representations or the encoder network by adding an additional prediction head on top, using four evaluation strategies: (1) Logistic regression (LR): A simple classifier is trained on the representations learned during pre-training; (2) $k$-nearest neightbors (kNN): The representations are evaluated using $k$-nearest neighbors, with $k$ set to match the number of shots $S$ for each task; (3) Linear Evaluation: The encoder network is frozen, and the representations are evaluated by training a single linear layer to predict the target labels for 100 epochs; (4) Fine-tuning: The encoder network is further trained with a few labeled samples to optimize the cross-entropy loss for 100 epochs. Each evaluation strategy is implemented using only a few labeled samples, and the resulting accuracy is assessed on the full labeled test datasets. As detailed in Section C.1, we found that the LR evaluation protocol consistently provides the best accuracy across various datasets and self-supervised algorithms. Therefore, in the main text, we report results primarily using the LR evaluation protocol.

**SubTab:** SubTab (Ucar et al., 2021) transforms tabular learning into a multi-view representation problem by dividing input features into multiple subsets. We follow the best configurations of the original paper, including the number of subset as 4 and batch size as 256. Detailed configuration can be found in `config/subtab.yaml`.

**SCARF:** SCARF (Bahri et al., 2021) is a contrastive learning framework that generates positive views by corrupting a random subset of features through sampling. We follow the best configurations of the original paper, including corruption rate as 0.6. Detailed configuration can be found in `config/scarf.yaml`.

**CL+Aug:** Following the setup of Bahri et al. (2021), we replace the augmentation modules to generate positive pairs during contrastive learning. We experiment with four augmentation methods:

masking, shuffling, noise, and RQ, using a selection ratio $p$ of 0.3 and a quantization scale (number of bins) of 10. Detailed configuration can be found in `config/ssl[AugName].yaml`.

**Binning:** Binning (Lee et al., 2024b) is a representation learning framework that predicts feature quantization bins instead of raw feature values, enhancing learning through reconstruction-based tasks. We follow the best configurations of the original paper, including the number of bins as 20. Detailed configuration can be found in `config/sslbinning.yaml`.

**Reconstruction:** We explore a simple reconstruction-based self-supervised learning method by predicting the raw feature values. The setup follows that of Lee et al. (2024b), with the only change being the objective function, defined as the mean squared error (MSE) between predicted and raw feature values. Detailed configuration can be found in `config/sslrecon.yaml`.

**VIME:** VIME (Yoon et al., 2020) introduces a self-supervised pretext task that involves estimating mask vectors from corrupted data, in addition to the reconstruction task. We follow the best configurations of the original paper, including the masking ratio as 0.3 and loss weights as 1. Detailed configuration can be found in `config/sslvime.yaml`.

**SAINT:** SAINT (Somepalli et al., 2021) uses attention over both rows and columns and employs augmentation techniques like CutMix in the input space and MixUp in the latent space. We follow the best configurations of the original paper, including CutMix ratio as 0.1 and hybrid attention. Detailed configuration can be found in `config/saint.yaml`.

### A.2.3 SEMI-SUPERVISED ALGORITHMS

In tabular semi-supervised learning (Yoon et al., 2020), a supervised loss is optimized with a few labeled samples, while an unsupervised loss is simultaneously optimized with unlabeled samples.

**VIME:** VIME (Yoon et al., 2020) defines a consistency loss as the mean squared error between original samples and their reconstructions from corrupted and masked samples with unlabeled samples. We follow the best configurations of the original paper, including the loss weight as 1 and learning steps as 1000. Detailed configuration can be found in `config/semivime.yaml`.

**Pseudolabels:** For unlabeled samples, pseudo-labels are often generated from original samples and used as supervised targets for augmented samples (Lee et al., 2013). We implement various tabular augmentation methods to generate these augmented samples. We use a default 2-layer MLP network as the classifier, and the detailed configuration for each augmentation can be found in `config/pseudolabel-[AugName].yaml`.

**Mean Teacher:** Mean Teacher (Tarvainen & Valpola, 2017) is semi-supervised learning method which uses the consistency loss between the teacher output and student output. The teacher model weights are updated as an exponential moving average of the student weights. We use a default 2-layer MLP network as the classifier, and the detailed configuration can be found in `config/meanteacher.yaml`.

**Interpolation Consistency Training (ICT):** ICT (Verma et al., 2022) is a semi-supervised learning method uses mean teacher framework while student parameters are updated to encourage the consistency between the output of mixed samples and the mixed output of the samples. We use the default 2-layer MLP network as the classifier, and the detailed configuration can be found in `config/ict.yaml`.

**Auto-encoders:** Auto-encoders use a reconstruction loss as the unsupervised regularization during training. We use the default 2-layer MLP network as the classifier, and the detailed configuration can be found in `config/ae.yaml`.

### A.2.4 FOUNDATION MODELS

Recent efforts in tabular domains have focused on developing foundation models trained on large-scale synthetic or real-world datasets.

**TabPFN:** TabPFN (Hollmann et al., 2022) is a Prior-Data Fitted Network (PFN) trained offline on synthetic datasets drawn from a prior that incorporates ideas from causal reasoning and favors simple structural causal models. However, TabPFN is limited to small tabular datasets, specifically those with fewer than 1000 training examples, 100 features, and 10 classes. For inference, a small set of labeled samples is required to construct the attention map for the specific dataset. We utilize the pretrained model weights and fit only the attention map during inference.

**HyperFast:** HyperFast (Bonet et al., 2024) is a hypernetwork designed for efficient classification of tabular data, capable of handling large-scale datasets. For pretraining, HyperFast utilize 70 real-world tabular datasets from OpenML library. During inference, labeled samples are used to generate dataset-specific target network weights. We utilize the pretrained model weights to produce these dataset-specific weights for accurate inference.

### A.2.5 UNSUPERVISED META-LEARNING ALGORITHMS

**STUNT:** STUNT (Nam et al., 2023b) generates diverse few-shot tasks by treating randomly chosen columns as target labels and employs a meta-learning scheme to learn generalizable knowledge through these constructed tasks. We follow the best configurations from the original paper, including setting the number of queries to 15 and using noise augmentation with a noise level of 0.1. Although STUNT allows the use of additional labeled datasets for validation, we do not utilize any additional labeled data during training. Detailed configurations can be found in `config/stunt.yaml`.

## A.3 AUGMENTATIONS

We provide the detailed descriptions for each augmentation methods suggested in the previous studies. For the hyperparameters of each method, we follow the best configuration reported in the original papers.

**Masking (Yoon et al., 2020; Huang et al., 2020)** : This method randomly masks a subset of feature values in the data by replacing them with a constant (typically zero). The hyperparameter is the selection ratio $p$, which determines the proportion of features to mask for each sample. In this study, we set the default selection ratio $p$ as 0.3.

**Sampling (Bahri et al., 2021)** : In the sampling approach, the selected feature values are replaced with values sampled from their empirical marginal distributions. This preserves the statistical properties of the original data but randomizes individual values. The hyperparameter is the selection ratio $p$, which controls the fraction of features to be replaced by sampled values. In this study, we set the default selection ratio $p$ as 0.3.

**Shuffling (Huang et al., 2020; Lee et al., 2024b)** : Shuffling involves randomly permuting the selected feature values within each feature column. The hyperparameter is the selection ratio $p$, which determines the proportion of feature values to be shuffled. In this study, we set the default selection ratio $p$ as 0.3.

**Noise (Nam et al., 2023b)** : Noise augmentation involves adding small random noise to a subset of feature values. The random noise is sampled from a Gaussian distribution with the mean as 0 and standard deviation of $\eta$. The hyperparameters are the selection ratio $p$ and noise level $\eta$. In this study, we set the default selection ratio $p$ as 0.3 and $\eta$ as 0.1.

**Subset (Ucar et al., 2021; Wang & Sun, 2022)** : The subset approach divides the input features into multiple subsets to generate different views of the data for multi-view representation learning. The hyperparameters is the number of subsets. In this study, we set this value as 4.

**CutMix (Somepalli et al., 2021)** : CutMix generates a new sample by combining two samples in the raw data space. A binary mask, determined by a combination ratio, specifies which features from the original sample are retained and which are replaced by corresponding features from a paired sample in the batch. The hyperparameter is the combination ratio. In this study, we set this value as 0.1.

**MixUp (Somepalli et al., 2021)** : MixUp augmentation linearly interpolates between a given sample and a randomly selected sample from the same batch in the embedding space. The hyperparameter is the combination ratio. In this study, we set this value as 0.2.

**Random quantization (Wu et al., 2023)** : Random quantization discretizes feature values by grouping them into bins, either uniformly or non-uniformly, and then sampling values randomly within each bin. The hyperparameter is the quantization scales, corresponding to the number of bins. In this study, we set this value as 10 per feature.

## B  RANGE-LIMITED AUGMENTATION

For a better understanding of our augmentation method, we provide a pseudo-code for implementation as follows.

---

**Algorithm 1** Self-Supervised Learning with Range-limited Augmentation

---

**Require:** Unlabeled dataset $D$, number of predefined ranges per feature $b$, probability of augmentation $p$, number of training epochs $T$, encoder network $f$, projection head $g$, optimizer
**Ensure:** Trained encoder $f$
1: **Define augmentation ranges**
2: **for** each feature $j$ in $D$ **do**
3:    Split the feature values into $b$ quantiles
4:    Define ranges $\mathbf{B}_j = \{B_{j,1}, B_{j,2}, \ldots, B_{j,b}\}$, where $B_{j,i} = (\beta_{j,i}^{\min}, \beta_{j,i}^{\max}]$ is the $i$-th range of the $j$-th feature
5: **end for**
6: **for** epoch $= 1$ to $T$ **do**
7:    **Sample mini-batch** of samples $\{x_k\}_{k=1}^{N}$ from $D$
8:    **Generate augmented views**
9:    **for** each sample $x_k$ in mini-batch **do**
10:      **for** each feature $j$ in $x_k$ **do**
11:        Draw a Bernoulli sample $m_{k,j} \sim \text{Bernoulli}(p)$
12:        **if** $m_{k,j} = 1$ **then**
13:          **Augment** $x_{k,j}$ using range-limited augmentation:
14:          **if** Shuffling mode **then**
15:            Shuffle values within the range $B_{j,i}$ containing $x_{k,j}$
16:          **else if** Sampling mode **then**
17:            Sample a new value from $\mathcal{U}(\beta_{j,i}^{\min}, \beta_{j,i}^{\max})$
18:          **end if**
19:        **end if**
20:      **end for**
21:    **end for**
22:    **Compute contrastive loss**
23:    Obtain representations $z_k = g(f(x_k))$ and augmented views $z_k' = g(f(x_k'))$
24:    Compute contrastive loss $\mathcal{L}_{\text{contrastive}}(z_k, z_k')$
25:    **Update parameters**
26:    Use optimizer to update parameters of $f$ and $g$ to minimize $\mathcal{L}_{\text{contrastive}}$
27: **end for**
28: **Return** trained encoder $f$

---

# C   ADDITIONAL RESULTS

In this study, we conduct extensive experiments on the FESTA benchmark, which includes 42 public datasets and 31 algorithms, evaluated over 50 random data splits and two different number of shots. This results in more than 100,000 scenarios tested based on accuracy, AUROC, and log loss. For clarity, we present the average accuracy, average ranks, and number of wins in the main text. Full results for individual scenarios are available in the FESTA benchmark, with a zip file included in the supplementary materials during the review process.

## C.1   FULL RESULTS FOR VARIOUS EVALUATION PROTOCOLS IN SELF-SUPERVISED ALGORITHMS

**Table 3:** Full results for various evaluation protocols in self-supervised algorithms: Once pre-training is completed, we evaluate the learned representations or the encoder network by adding an additional prediction head on top, using four evaluation strategies: (1) Logistic regression (LR): A simple classifier is trained on the representations learned during pre-training; (2) $k$-nearest neightbors (kNN): The representations are evaluated using $k$-nearest neighbors, with $k$ set to match the number of shots $S$ for each task; (3) Linear Evaluation: The encoder network is frozen, and the representations are evaluated by training a single linear layer to predict the target labels for 100 epochs; (4) Fine-tuning: The encoder network is further trained with a few labeled samples to optimize the cross-entropy loss for 100 epochs. Each evaluation strategy is implemented using only a few labeled samples, and the resulting accuracy is assessed on the full labeled test datasets. Due to the superior performance of LR across diverse datasets and algorithms, we report the accuracy with the LR prediction head in the main text.

| Model | Evaluation protocol | 1-shot accuracy (%) | 5-shot accuracy (%) |
|---|---|---|---|
| Reconstruction | LR | 33.414±16.978 | 32.816±17.381 |
| Binning | LR | 34.564±17.248 | 34.114±16.994 |
| VIME | LR | 35.999±17.520 | 36.428±18.166 |
| SubTab (CL+Subset) | LR | 36.264±17.614 | 36.680±18.005 |
| SCARF (CL+Sampling) | LR | 48.830±14.716 | 59.170±16.073 |
| SAINT (CL+CutMix+MixUp) | LR | 45.191±18.857 | 50.768±20.715 |
| CL+Masking | LR | 48.114±14.885 | 56.787±17.365 |
| CL+Shuffling | LR | 49.091±14.899 | 59.373±16.233 |
| CL+Noise | LR | 49.076±14.881 | 59.394±16.263 |
| CL+RQ | LR | 47.153±16.012 | 55.882±18.437 |
| CL+Range-limited Shuffling | LR | 51.972±15.243 | 61.921±16.641 |
| CL+Range-limited Sampling | LR | 50.640±14.759 | 60.647±16.315 |
| Reconstruction | kNN | 33.333±17.006 | 32.956±17.448 |
| Binning | kNN | 34.573±17.252 | 34.292±17.301 |
| VIME | kNN | 36.072±17.400 | 36.365±17.970 |
| SubTab (CL+Subset) | kNN | 36.205±17.647 | 36.482±17.867 |
| SCARF (CL+Sampling) | kNN | 48.489±14.990 | 53.177±16.507 |
| SAINT (CL+CutMix+MixUp) | kNN | 45.592±18.562 | 48.992±19.947 |
| CL+Masking | kNN | 48.118±14.734 | 53.053±16.532 |
| CL+Shuffling | kNN | 48.781±15.060 | 53.558±16.449 |
| CL+Noise | kNN | 48.819±14.965 | 53.628±16.448 |
| CL+RQ | kNN | 47.314±15.806 | 51.906±17.335 |
| CL+Range-limited Shuffling | kNN | 50.188±14.683 | 55.387±16.438 |
| CL+Range-limited Sampling | kNN | 49.938±14.548 | 55.250±16.339 |
| Reconstruction | Linear evaluation | 32.081±17.428 | 32.354±17.347 |
| Binning | Linear evaluation | 32.222±17.391 | 32.226±17.426 |
| VIME | Linear evaluation | 32.106±17.455 | 32.075±17.455 |
| SubTab (CL+Subset) | Linear evaluation | 32.031±17.519 | 32.086±17.499 |
| SCARF (CL+Sampling) | Linear evaluation | 36.550±17.880 | 36.431±17.787 |
| SAINT (CL+CutMix+MixUp) | Linear evaluation | 36.821±17.891 | 36.843±17.890 |
| CL+Masking | Linear evaluation | 36.647±17.991 | 36.722±17.914 |
| CL+Shuffling | Linear evaluation | 36.766±18.211 | 36.728±18.026 |
| CL+Noise | Linear evaluation | 36.498±17.762 | 36.759±18.056 |
| CL+RQ | Linear evaluation | 36.502±17.934 | 36.352±17.881 |
| CL+Range-limited Shuffling | Linear evaluation | 36.699±17.957 | 36.514±17.949 |
| CL+Range-limited Sampling | Linear evaluation | 36.262±17.815 | 36.627±17.867 |
| Reconstruction | Fine-tuning | 32.127±17.454 | 32.232±17.402 |
| Binning | Fine-tuning | 32.182±17.442 | 32.186±17.460 |
| VIME | Fine-tuning | 31.947±17.558 | 31.981±17.537 |
| SubTab (CL+Subset) | Fine-tuning | 32.402±17.311 | 32.462±17.293 |
| SCARF (CL+Sampling) | Fine-tuning | 36.938±18.309 | 36.815±18.205 |
| SAINT (CL+CutMix+MixUp) | Fine-tuning | 36.441±17.857 | 36.506±17.878 |
| CL+Masking | Fine-tuning | 36.788±18.121 | 36.794±18.117 |
| CL+Shuffling | Fine-tuning | 36.858±18.026 | 36.719±18.074 |
| CL+Noise | Fine-tuning | 36.865±18.313 | 36.747±18.032 |
| CL+RQ | Fine-tuning | 37.023±18.360 | 37.214±18.222 |
| CL+Range-limited Shuffling | Fine-tuning | 36.989±18.137 | 36.701±18.234 |
| CL+Range-limited Sampling | Fine-tuning | 36.852±18.169 | 36.841±18.212 |

## C.2 COMPUTATIONAL EFFICIENCY ON TRAINING TIME

**Table 4:** We provide the average training time for each algorithm, all implemented on a single NVIDIA GeForce RTX 3090. While our approach incurs a slight increase in training time due to the overhead from range-limited augmentations, this increase is minimal compared to the significant performance improvements observed. Moreover, our approach remains efficient even when compared to more complex architectures like transformers. Notably, the training time does not scale directly with increasing $b$, indicating that the choice of $b$ has a limited effect on computational cost. We also note that inference time remains unaffected as long as the classifier architecture is unchanged.

| Shots | | 1 | | | 5 | | |
|---|---|---|---|---|---|---|---|
| Model | | Fitting time per epoch (secs) | Epochs | Fitting Time (secs) | Fitting time per epoch (secs) | Epochs | Fitting Time (secs) |
| Supervised | LR | - | - | 0.006±0.009 | - | - | 0.011±0.034 |
| | kNN | - | - | 0.002±0.003 | - | - | 0.002±0.002 |
| | XGBoost | - | - | 0.531±0.397 | - | - | 0.547±0.645 |
| | CatBoost | - | - | 15.789±89.569 | - | - | 29.178±106.585 |
| | LightGBM | - | - | 1.703±9.417 | - | - | 6.304±36.666 |
| | MLP | 0.013 | 100 | 1.301±0.078 | 0.014 | 100 | 1.392±0.212 |
| Semi-Supervised | VIME | - | 1000 steps | 14.733±7.297 | - | 1000 steps | 14.760±6.932 |
| | AE | 1.521 | 100 | 152.064±347.450 | 1.547 | 100 | 154.710±348.220 |
| | ICT | 0.215 | 100 | 21.521±46.982 | 0.226 | 100 | 22.571±44.978 |
| | MeanTeacher | 0.537 | 100 | 53.671±131.578 | 0.490 | 100 | 48.950±113.615 |
| | PL+Masking | 1.092 | 20 | 21.847±49.487 | 1.138 | 20 | 22.749±50.598 |
| | PL+Sampling | 1.108 | 20 | 22.158±50.377 | 1.118 | 20 | 22.353±49.625 |
| | PL+Shuffling | 1.712 | 20 | 34.245±74.334 | 1.734 | 20 | 34.670±75.218 |
| | PL+Noise | 1.716 | 20 | 34.310±75.265 | 1.727 | 20 | 34.534±74.287 |
| | PL+RQ | 1.333 | 20 | 26.652±60.823 | 1.400 | 20 | 27.854±63.031 |
| | PL+CutMix | 1.375 | 20 | 27.500±63.558 | 1.417 | 20 | 28.337±64.607 |
| Unsup. Meta | STUNT | - | 10000 steps (Early stop) | 16.842±45.128 | - | 10000 steps (Early stop) | 12.907±25.712 |
| Foundation | HyperFast | - | - | 29.837±2.246 | - | - | 30.553±2.591 |
| Self-supervised | Reconstruction | 1.757 | 23.619±13.703 | 41.366±115.109 | 1.742 | 23.000±12.949 | 40.057±103.712 |
| | Binning | 1.813 | 23.738±13.791 | 43.044±96.760 | 1.710 | 25.190±14.484 | 43.071±107.028 |
| | VIME | 1.572 | 10 | 15.719±27.034 | 1.303 | 10 | 13.028±24.221 |
| | SubTab (CL+Subset) | 0.783 | 20 | 15.659±30.695 | 0.770 | 20 | 15.385±29.777 |
| | SCARF (CL+Sampling) | 1.692 | 12.667±6.038 | 21.428±51.663 | 1.596 | 13.976±6.816 | 22.310±62.295 |
| | SAINT (CL+CutMix+MixUp) | 5.363 | 50 | 268.140±604.555 | 5.300 | 50 | 264.983±616.131 |
| | CL+Masking | 0.942 | 19.310±11.081 | 18.183±36.874 | 0.921 | 17.381±8.258 | 16.000±35.271 |
| | CL+Shuffling | 2.094 | 19.619±8.856 | 41.092±111.074 | 2.117 | 21.381±10.305 | 45.271±116.111 |
| | CL+Noise | 2.139 | 18.595±11.350 | 39.779±106.031 | 2.481 | 19.762±9.961 | 49.037±131.432 |
| | CL+RQ | 1.925 | 6.643±2.959 | 12.786±30.661 | 1.296 | 8.048±5.912 | 10.428±18.567 |
| | CL+Range-limited Shuffling | 4.685 | 23.333±9.511 | 109.311±287.777 | 5.128 | 23.905±10.251 | 122.580±331.106 |
| | CL+Range-limited Shuffling ($b = 2$) | 3.429 | 24.571±10.821 | 84.260±188.626 | 4.572 | 22.714±9.733 | 103.855±290.444 |
| | CL+Range-limited Sampling | 5.187 | 22.524±8.889 | 116.823±303.076 | 6.189 | 26.405±9.976 | 163.426±419.510 |
| | CL+Range-limited Sampling ($b = 2$) | 6.056 | 24.881±10.425 | 150.684±458.117 | 6.063 | 24.333±9.125 | 147.537±391.090 |
| Self-supervised | CL+Range-limited Shuffling† | 4.321 | 22.576±10.234 | 97.554±228.756 | 4.874 | 24.727±11.263 | 120.517±307.538 |
| | CL+Range-limited Sampling† | 5.880 | 24.212±10.710 | 142.355±359.792 | 6.422 | 24.515±12.481 | 157.434±417.885 |
| Foundation | TabPFN† | - | - | 0.001±0.000 | - | - | 0.001±0.000 |

