# OpenReview forum: "Range-limited Augmentation for Few-shot Learning in Tabular Data"
_ICLR.cc/2025/Conference — Submitted to ICLR 2025_

### Official Review · Reviewer_NQQr · 2024-10-28

**Soundness:** 3
**Presentation:** 3
**Contribution:** 3
**Rating:** 5
**Confidence:** 3

**Summary:**

This paper introduces the limited range expansion of a small amount of learning in tabular data. This method aims at maintaining semantic consistency in the learning process, which is very important to improve the classification performance under the condition of limited labeled data. The authors have proved the effectiveness of their method through a new benchmark test, which is superior to the existing methods.

**Strengths:**

Originality: The concept of range-limited augmentation for contrastive learning in tabular data is novel. The paper introduces a creative solution to the challenge of designing augmentations that preserve task-relevant information in a domain. The application of distribution-based range limiting to augmentation strategies is an innovative approach that leverages the inherent structure of tabular data, which is a creative combination of existing ideas in data augmentation and representation learning. The introduction of the FESTA benchmark is original as it addresses a gap in the evaluation of few-shot learning methods for tabular data, providing a new resource for the community.
Quality: The paper is well-structured, with a clear problem statement, detailed methodology, comprehensive experiments. The paper provides a rigorous evaluation of the proposed method through extensive experiments on a diverse set of datasets and algorithms.
Clarity: The paper is well-written, with a clear and logical flow that makes it easy to follow the authors' line of reasoning.
Significance: The paper addresses a significant problem in the field of machine learning, where labeled data is often scarce, especially in specialized domains like healthcare or finance. The proposed solution has the potential to impact a wide range of applications by enabling more effective learning from limited labeled data, which is a common scenario in many real-world scenarios. It provides a standardized way to evaluate and compare few-shot learning methods in tabular data, which could drive further research and development in this area.

**Weaknesses:**

1. The paper provides empirical evidence of the effectiveness of the proposed method but does not offer a theoretical analysis of why range-limited augmentation works better than other methods in preserving task-relevant information.  Developing a theoretical framework to explain these observations could further strengthen the contribution.
2. In classifiation-related papers, confusion matrix is an important basis for judging a classifier, so it is suggested to add more blanks here.
3. There are some typos in the manuscripts. The presentation of this manuscript can be improved.

**Questions:**

1. Can you provide a theoretical explanation for how effective a scope-limited augmentation method is in preserving task-related information compared to other augmentation methods?
2. What specific optimizations can be implemented to reduce the computational overhead associated with scope-constrained enrichment?
3. Will the benchmark code be exposed on a platform like GitHub to promote reproducibility?
4. Based on the findings of this study, what are the plans for future research?

---

> ### Author Response · Authors · 2024-11-20
>
> - **[Lack of theoretical analysis]** Thank you for raising this important point. While a theoretical explanation would indeed strengthen the contribution of our work, defining task-relevant information or assuming that a specific augmentation preserves semantic information is far from trivial in tabular domains, unlike in image or text data. As mentioned in the manuscript (L47-53), assessing task labels for tabular data is challenging and often infeasible without domain expertise.
>
>     Instead of relying on potentially unrealistic definitions or assumptions about ground-truth labeling functions, we adopted two metrics — usable information and representation invariance score — to evaluate the preservation of semantic information, using a reliable proxy for the labeling function. In the revision, we will add theoretical implications derived from our empirical analysis in Section 4.1, including the relationships between these metrics and how they reflect the preservation of semantic information. As referenced in prior studies, these metrics provide valuable insights into the efficacy of augmentations in retaining task-relevant information.
>
> - **[Lack of confusion matrix based metrics]** As noted in the manuscript (L211), we evaluate not only accuracy but also AUROC and log loss in our benchmark, with results provided in the supplementary materials. Specifically, our method achieved an average AUROC of 0.653 in the 1-shot setup and 0.734 in the 5-shot setup, consistently outperforming other approaches. We appreciate the reviewer’s suggestion to include additional results, such as confusion matrices, to provide a more comprehensive evaluation of our method’s effectiveness. We will incorporate these results into the revised manuscript to enhance clarity and support our findings.
> - **[Typos]** Thank you for your feedback. We will carefully proofread the manuscript to address any remaining typographical errors and improve the overall presentation. If the reviewer has identified specific issues, we would greatly appreciate further guidance to ensure these are thoroughly corrected.
> - **[Computational overhead]** Thank you for raising this issue. As noted in the manuscript (L259) and Section 6.2, our method introduces computational overhead due to the need for sampling or shuffling within individual ranges. However, as summarized in Supplementary C.2, when comparing fitting time per epoch (excluding the impact of early stopping), our method exhibits lower training time than transformer-based approaches like SAINT, as it relies on a simple MLP architecture. Despite this simplicity, our augmentation method consistently enhances few-shot learning performance across diverse datasets.
>
>     We acknowledge that the computational overhead could present challenges for practical applications. As a potential optimization, leveraging the distributional properties of the ranges (which contain an equal number of samples) could allow precomputing shuffled or sampled values for each range, reducing runtime by reusing indexed augmented samples. We will explore and implement this optimization in the provided code to improve efficiency further.
>
> - **[Benchmark code availability]** Absolutely. The reproducible benchmark is already included in the supplementary materials and will be made publicly available through a GitHub repository following publication.
> - **[Future research plans]** Thank you for your interest in our research. Augmentations are not only essential for contrastive learning but can also serve as powerful tools for supervised learning and data amplification in scarce data scenarios, such as anomaly detection. Building on our findings that range-limited augmentations effectively preserve semantic information in tabular datasets and enhance few-shot learning performance, we plan to extend this approach to other applications, including supervised learning tasks and anomaly detection systems.

---

> > ### Comment · Reviewer_NQQr · 2024-12-03
> >
> > Thank you for your refutation and revision. I have read them all carefully, and I still maintain my original score.

---

### Official Review · Reviewer_ksSf · 2024-10-31

**Soundness:** 2
**Presentation:** 2
**Contribution:** 2
**Rating:** 5
**Confidence:** 5

**Summary:**

This paper addresses a challenge in tabular data with approaches from few-shot learning. In particular, this paper proposes range-limited augmentation equipped with contrastive learning to build a model capable of tackling learning on tabular domains. The proposed approach provides augmentation by shuffling or sampling values within predefined feature-specific ranges, preserving semantic consistency. In addition to the proposed approach, this paper also introduces a benchmark to evaluate the efficacy of  the proposed approach compared to the counterparts so-called Few-Shot Tabular Classification (FESTA). Through experiments on the benchmark, the proposed approach with augmentation with contrastive learning the performance can be improved compared to the state-of-the-arts in few-shot learning.

**Strengths:**

- This paper introduces an underexplored problem by focusing few-shot learning on tabular data. This is due to data unavailability in tabular domains, prohibiting training in a large scale manner. Thus, few-shot learning is a reasonable perspective in tackling such problems.
- This work also introduces a novel benchmark with 42 tabular datasets and 31 techniques for comparison. This benchmark has the potential to serve as a resource for future research efforts, providing a standardized comparison and facilitating the evaluation of new methods in tabular data analysis.

**Weaknesses:**

- The problem scope is quite narrow and the proposed approach only works well in tabular domains. Additionally, the proposed augmentation approach demonstrates effectiveness primarily within tabular data domains and might not generalize to other types of data structures or formats.
- This paper has serious clarity issues in terms of problem introduction. The problem in this work is not elaborated well in the paper. The problem setup is quite general in which previous few-shot learning papers have discussed it. However, a specific focus problem setup on tabular domains is not described in the paper. Additionally, there is no clear problem description to differentiate among few-shot learning in several domains e.g., image, text, and tabular domains. This causes confusion in understanding the core problem in tabular domains compared to other modalities.
- Based on my understanding, the augmentation method is based on the ranged limited shuffling and sampling. However, this method is largely inspired by augmentation methods in the image domain, but here it is applied to a different domain. The strategy closely resembles what SimCLR introduced in its paper, with only a specific range constraint added to the technique. In SimCLR, we can interpret “range-limited sampling” as modifying the colors from a range between 0-255, and shuffling as cropping from a specific region. Therefore, the novelty of this approach sounds quite limited.
- The writing is quite poor. There are many irrelevant points in the paper e.g., lists of detailed baselines and augmentation. These sections do not provide meaningful insights or support the main contributions. As a result, the core contributions are obscured by unrelated information that does not directly connect to the main focus of the paper.

**Questions:**

This paper missed explanation on how each method can be used to the problem at hands. For instance, XGBoost, CatBoost, LightBGM are not methods designed for few-shot learning. How are these methods modified to accommodate learning with a few data?

---

> ### Author Response · Authors · 2024-11-20
>
> - **[Generalizability to other data domains]** We respectfully disagree with the reviewer’s assessment. In recent years, a substantial number of studies have been dedicated to tabular learning, underscoring its importance in real-world applications. Tabular data is ubiquitous across domains such as healthcare, finance, and education, where high-stakes decisions rely on robust learning models.
>
>     Unlike image or text data, tabular data presents unique challenges, including heterogeneous feature types (numerical and categorical) and lack of spatial or sequential structure. Consequently, standard augmentation methods from other domains, such as random rotation or flipping in images, are not directly applicable to tabular data. Our approach addresses these challenges with a tailored solution that effectively preserves task-relevant semantics within tabular datasets. We strongly believe that tabular-data-specific approaches are neither narrow nor trivial but rather essential to advancing machine learning in this domain.
>
> - **[Clarity issues in problem introduction]** Thank you for raising this point. While we believe the current manuscript provides sufficient context to understand our target problem, we agree that additional clarity on the unique challenges of tabular data would be beneficial. In the revision, we will include a detailed discussion of the tabular-specific problem setup, including the diverse feature dimensionality, mixed feature types (categorical and numerical), and the lack of domain-specific inductive biases like spatial or temporal structure. We hope these revisions will further enhance the accessibility and clarity of our work for all readers.
> - **[Limited novelty compared to SimCLR)]** We respectfully disagree and believe the reviewer may have misunderstood our method. As noted in our official comment, other reviewers recognized the novelty of our range-limited augmentation technique, which preserves task-relevant information within feature-specific ranges — a critical requirement for tabular data.
>
>     To clarify the misunderstanding, let us address the specific examples provided in the review. The color jittering in SimCLR is similar to the sampling method proposed in SCARF, while our range-limited sampling applies sampling within feature-specific ranges, tailored to tabular data. Similarly, random cropping in SimCLR resembles the subset method used in SubTab but is entirely different from our approach. Our range-limited shuffling is unique to tabular domains, where values for a given feature are shuffled only within predefined ranges, preserving semantic consistency. While shuffling might loosely resemble permutation augmentation in image domains, it is fundamentally adapted for the unique structure of tabular data. As noted in the manuscript, the novelty of our method lies in introducing range constraints to well-known augmentation techniques, significantly enhancing semantic preservation in tabular datasets.
>
>     We hope this explanation clarifies the novelty of our augmentation method and its contribution to contrastive learning in tabular domains.
>
> - **[Writing quality]** We respectfully disagree with the reviewer’s point. One of the main contributions of our study is the introduction of FESTA, a new benchmark for few-shot learning in tabular domains. To establish the novelty of this benchmark, it is essential to explain the datasets and algorithms included and to provide a rigorous comparison. Furthermore, as there has been no prior systematic investigation of few-shot learning in tabular domains, detailing the experimental setup is crucial to ensure reproducibility and clarity. Other reviewers have noted that the paper is well-organized, with clear problem definitions, methods, and extensive evaluations across diverse datasets, making the contributions easy to follow.
>
>     If the reviewer identifies specific sections as unnecessary or unclear, we would appreciate further clarification and will gladly revise or condense these sections as needed.
>
> - **[Use of GBDT methods]** Thank you for raising this question. While it is true that gradient-boosted decision trees (GBDTs) are not inherently few-shot learning algorithms, they are widely recognized for their strong performance in tabular data tasks, often surpassing deep neural networks in many setups. Including GBDT methods in our evaluation allows for a fair comparison and aligns with practices in prior studies, such as STUNT, which used similar setups.
>
>     In our experiments, we followed the same task setups as those used in previous works, including STUNT, and included detailed configuration descriptions in the supplementary material. Additionally, the provided code includes all configurations and executable scripts to ensure reproducibility. If further explanation is needed, we can expand the supplementary material to describe how these methods were adapted for few-shot scenarios.

---

> > ### Comment · Reviewer_ksSf · 2024-11-26
> > **Feedback**
> >
> > Thanks for providing the rebuttal and revision. Though the paper has some potentials, The idea, contributions, novelties in conceptual comparison with the existing techniques are lacking in its current form. I agree that tabular data is a challenging and critical problems in the learning methods. But, the way of explaining the proposed method and the problem correlated to the problem at hands (i.e., few-shot learning for tabular data) sounds detached. Furthermore, the revised version and rebuttal do not address well my concerns. I would keep my current rating.

---

### Official Review · Reviewer_ceaX · 2024-11-02

**Soundness:** 2
**Presentation:** 2
**Contribution:** 2
**Rating:** 3
**Confidence:** 5

**Summary:**

The paper proposed two things. First, the few-shot learning benchmark for tabular datasets with a focus on 1 and 5-shot learning. Second, they proposed a data augmentation method for tabular datasets by additionally considering the range of the feature value for the feature augmentation. The author shows that the proposed method is effective by combining it with contrastive learning.

**Strengths:**

1. The overall paper is well-written and easy to read.

2. The author has conducted a new benchmark.

**Weaknesses:**

1. The proposed method is only applicable to numerical values. This is a highly critical weakness since many tabular datasets consist of categorical values.

2. Limited novelty. The proposed idea is a data augmentation method with a range limitation. In this regard, the only novelty is the range limitation, which I think is sensible as an engineering technique yet not as novel as an academic paper to be accepted.

3. The focused problem is too narrow. I do believe few-shot learning for tabular datasets [1] is important, but the more important problem is the full-shot setup. I think in a full-shot setup, the most dominant methods are tree-based methods, where the proposed method is somewhat hard to utilize. Furthermore, I think the author should consider a broader range of few-shot if they want to focus on few-shot learning, e.g., 8, 16, 32, and 64-shot learning.

4. In a few-shot tabular learning setup, recent LLM-based methods have become effective. I think it would be interesting to combine the current method with the LLM-based, few-shot, tabular learning methods [2,3,4,5].

5. It would be more interesting to suggest a new self-supervised learning (or any learning) method that can better leverage the proposed augmentation. I believe this will enhance the novelty.

In summary, I recommend rejection. The weakness mentioned above is quite critical and needs to be addressed.

Reference\
[1] STUNT: Few-shot Tabular Learning with Self-generated Tasks from Unlabeled Tables, ICLR 2023\
[2] LIFT: Language-Interfaced Fine-Tuning for Non-Language Machine Learning Tasks, NeurIPS 2022\
[3] Language models are weak learners, NeurIPS 2023\
[4] Large Language Models Can Automatically Engineer Features for Few-Shot Tabular Learning, ICML 2024\
[5] Tabular Transfer Learning via Prompting LLMs, COLM 2024

**Questions:**

Please refer to the weakness part above.

---

> ### Author Response · Authors · 2024-11-20
>
> - **[Adaptation to categorical features]** Thank you for your comment. As noted in the manuscript (L483), the current method is specifically designed for numerical features, and its extension to categorical data remains a limitation. Categorical features typically have limited discrete values, which can lead to unintended semantic changes during augmentation. (In this study, as outlined in L182, we followed the definition from [Lee et al., 2024b] to classify categorical features as those with fewer than 20 unique values.) While range-limited augmentations for numerical features, such as height (e.g., 140–160 cm, 161–180 cm), preserve task-relevant semantics, similar operations on categorical features like nationality may result in unrealistic or misleading variations. Therefore, we intentionally excluded categorical features from augmentation and will clarify this design choice in the revised manuscript.
> - **[Limited novelty]** We understand the reviewer’s concern regarding the perceived novelty of our range-limited augmentation method, as it builds upon existing sampling and shuffling techniques. However, as other reviewers (NQQr, qfF3) have noted, our approach introduces a significant advancement in contrastive learning for tabular data by preserving task-relevant information more effectively through range-limited augmentation. While the concept may seem straightforward, its impact is far from negligible, as evidenced by its consistent performance improvements.
>
>     Moreover, we believe our work provides meaningful contributions beyond the augmentation method itself: (1) introducing a novel augmentation technique tailored specifically to tabular data, (2) presenting FESTA, the first comprehensive benchmark for few-shot tabular learning, and (3) providing an empirical analysis of semantic preservation in different augmentation methods. Together, these contributions advance the understanding and practical capabilities of few-shot learning in tabular domains.
>
> - **[Importance of few-shot learning tasks]** We acknowledge the importance of full-shot tabular learning, which has been extensively investigated in prior benchmark studies, including WhyTrees [1], OpenML, TabZilla [D McElfresh et al., 2023], and TabRepo [D Salinas et al., 2023], where tree-based models often dominate on average performance. These benchmarks provide a solid foundation for evaluating full-shot learning, and substantial progress has already been made in this area.
>
>     In contrast, few-shot learning in tabular data remains underexplored, despite its critical importance in domains constrained by privacy limitations and the high cost of labeling, which often lead to data scarcity. As noted by other reviewers (ksSf, NQQr), few-shot learning directly addresses these challenges, making it an essential and practical research topic in tabular data applications.
>
>     Additionally, we have already presented results for a range of shot settings in Section 6.1, showing that our method consistently achieves robust performance gains across various numbers of shots. This underscores the practical value and adaptability of our approach to real-world scenarios with limited labeled data.
>
>     [1] Léo Grinsztajn et al., Why do tree-based models still outperform deep learning on tabular data?, NeurIPS, 2022.
>
> - **[LLM-based baselines]** We agree that LLM-based methods hold significant potential for few-shot tabular learning and have cited relevant studies in Section 2 (L127). However, as explained in the manuscript (L129), our study intentionally focuses on approaches that do not rely on auxiliary inputs, such as column names or descriptions. These inputs are often incomplete, inconsistent, or unavailable in real-world scenarios, which limits the practicality of LLM-based approaches in certain domains.
>
>     By excluding methods that depend on external metadata, we aim to provide a fairer comparison among approaches that operate solely on the data itself. Nonetheless, we recognize the potential for integrating range-limited augmentation with LLM-based methods and will explore this direction in future work.
>
> - **[Suggesting a new SSL framework]** Thank you for your suggestion. As noted in the manuscript (L256), we follow the contrastive learning framework proposed in SCARF. This deliberate choice was made to isolate and highlight the impact of preserving semantic information through augmentation methods and to demonstrate how this preservation contributes to improving few-shot learning performance. By keeping the overall framework consistent, we aim to emphasize the novelty and effectiveness of our range-limited augmentation approach.
>
>     That said, we acknowledge the value of developing a new learning framework to further leverage the proposed augmentation method. If the reviewer feels that the current focus on augmentation limits the perceived novelty, we will consider extending the framework as part of our future work.

---

> > ### Comment · Reviewer_ceaX · 2024-11-22
> > **Thank you for the rebuttal**
> >
> > Thank you for taking the time to respond to my question during the rebuttal. However, after reading the rebuttal, I found there were no misunderstandings from my original review, and I still believe the weakness holds. I do agree that the benchmark is indeed an important contribution, however, the main contribution is the method itself (as noted in the title). I think the limited novelty is also mentioned by reviewer ksSf. Also, I still think the mentioned limitation (i.e., can't be used for categorical feature) is a critical issue typically for tabular data, thereby I will keep my original score. Thank you again for the rebuttal.

---

### Official Review · Reviewer_qfF3 · 2024-11-03

**Soundness:** 2
**Presentation:** 2
**Contribution:** 2
**Rating:** 6
**Confidence:** 3

**Summary:**

The paper presents a novel data augmentation method called Range-Limited Augmentation, which is designed to improve few-shot learning for tabular data. The proposed approach involves shuffling or sampling feature values within predefined ranges to maintain semantic consistency and improve model performance. Also, the paper introduces a Few-Shot Tabular Classification Benchmark (FESTA), encompassing 42 tabular datasets and 31 algorithms to evaluate the effectiveness of their approach. The paper demonstrates that the proposed method achieves superior results in one-shot classification scenarios, outperforming existing supervised, unsupervised, and foundation models.

**Strengths:**

1. The proposed range-limited augmentation is a unique solution to the challenge of maintaining semantic consistency in tabular data augmentations, which is crucial for effective contrastive learning.
2. The proposed FESTA benchmark offers a large-scale, diverse set of tabular datasets and learning paradigms, making it a valuable for future research opening.
3. The method shows substantial performance improvements, especially in 1-shot scenarios, and maintains competitive performance in 5-shot setups.
4. The analysis using mutual information and representation invariance score seems a reasonable choice to evaluate the quality of data augmentation.

**Weaknesses:**

1. The proposed method seems straightforward when applying to numerical features, but it is unclear how it can be adapted to categorical data. Also, it is not clearly analyzed how the proposed method perform in homogeneous tabular dataset.
2. The proposed method is evaluated only on proposed FESTA benchmark, which may not cover the entire trend in tabular data learning. For example, the paper could include OpenML-CC18 dataset as preliminary benchmark.
3. The proposed method only considers classification task, where regression tasks are also important aspect of tabular data learning.

**Questions:**

1. Could the author elaborate how the proposed range limited shuffling and range limited sampling applies to categorical features?
2. Could the author provide empirical results on other popular OpenML dataset?
3. Could the proposed method be applied to regression tasks?

---

> ### Author Response · Authors · 2024-11-20
>
> - **[Adaptation to categorical features]** Thank you for your insightful comment. As noted in the manuscript (L483), the current method is specifically designed for numerical features, and its extension to categorical data remains a limitation. Categorical features typically have limited discrete values, which can lead to unintended semantic changes during augmentation. (In this study, as outlined in L182, we followed the definition from [Lee et al., 2024b] to classify categorical features as those with fewer than 20 unique values.) While range-limited augmentations for numerical features, such as height (e.g., 140–160 cm, 161–180 cm), preserve task-relevant semantics, similar operations on categorical features like nationality may result in unrealistic or misleading variations. Therefore, we intentionally excluded categorical features from augmentation and will clarify this design choice in the revised manuscript.
>
>     Regarding homogeneous datasets, our method is applicable to datasets comprising only numerical features. In the FESTA benchmark, we identified 16 datasets (IDs: 22, 1475, 1489, 1492, 1497, 4153, 40499, 44125, 44131, 1462, 44126, 41168, 44091, 44123, 44090, 40922) that fall into this category. Across these datasets, our method consistently outperformed competing approaches, achieving an average accuracy of 0.534 with average rank of 2.125 in the 1-shot setup (second-best: MLP — Accuracy 0.504, Rank 9.813), and 0.656 with average rank of 2.625 in the 5-shot setup (second-best: MLP — Accuracy 0.640, Rank 6.938).
>
> - **[Evaluation on other benchmarks]** We appreciate this valuable suggestion. To the best of our knowledge, there has been no large-scale benchmark explicitly designed for few-shot learning in tabular data, despite its importance in domains where privacy and cost constraints often limit data availability. To address this gap, we introduced FESTA, the first comprehensive benchmark tailored to this scenario, comprising 42 dataset systematically selected based on criteria summarized in Section 3.2.
>
>     While OpenML-CC18 is a well-known collection, it is not suitable for evaluating few-shot learning algorithms due to its inclusion of datasets that violate some necessary conditions. For instance, some datasets have fewer samples per class than the number of shots (5 in our setup), while others correspond to flattened image datasets, such as MNIST or CIFAR-10, which are not representative of the unique characteristics of tabular data. To address these limitations of existing tabular benchmarks, we built the FESTA benchmark to evaluate few-shot learning algorithms in tabular domains while enhancing generalizability.
>
>     That said, we acknowledge the importance of benchmark diversity in enhancing generalizability.  To address this concern, we conducted additional experiments on 29 tabular datasets from OpenML-CC18 that meet our selection criteria. Our method (CL+BinShuffling) consistently demonstrated superior performance, achieving an average 1-shot accuracy of 0.523 and AUROC of 0.723, compared to the second-best methods, CL+Shuffling (accuracy: 0.518) and SCARF (AUROC: 0.716). We will incorporate these results into the revised manuscript and include additional datasets, such as credit-g, to further validate the generalizability of our findings.
>
> - **[Application to regression tasks]** As detailed in Section 6.3, we have already extended our analysis to include few-shot regression tasks and demonstrated that the proposed method performs consistently well. In specific, our range-limited augmentation methods (shuffling and sampling) achieved superior performance compared to strong baselines like XGBoost and MLP across multiple regression datasets.

---

> > ### Comment · Reviewer_qfF3 · 2024-11-26
> > **Post-rebuttal**
> >
> > I appreciate the authors' efforts for the rebuttal. I raised my score to 6 as most of my concerns are resolved.

---

### Author Response · Authors · 2024-11-20

Dear Reviewers,

We would like to express our sincere gratitude to the reviewers for their thoughtful and constructive feedback on our manuscript. We greatly appreciate the time and effort each reviewer has taken to engage with our work, and we welcome further discussions that can help improve the quality of the manuscript.

Here, we summarize the strengths of our paper as highlighted in the reviews:

1. Novel contributions
    - Range-Limited Augmentation: A novel technique for contrastive learning in tabular data, preserving task-relevant information within feature-specific ranges. (NQQr, qfF3)
    - FeSTa Benchmark: A benchmark for few-shot learning on tabular data, including diverse datasets and methods, supporting standard evaluation and future research. (NQQr, qfF3, ceaX, ksSf)
2. Addressing a Key Challenge
    - Few-Shot Learning in Tabular Domains: Targeting the crucial issue of data scarcity in tabular fields like healthcare and finance, where privacy and cost limit data availability. (ksSf, NQQr)
3. Methodological Rigor and Performance
    - Robust Evaluation Metrics and Results: The proposed method demonstrates substantial performance gains, particularly in 1-shot scenarios, while remaining competitive in 5-shot setups. This effectiveness is further supported by empirical analysis for the quality of data augmentation (qfF3).
4. Clarity and Presentation
    - Clear and Structured: Well-organized, with clear problem definition, methods, and extensive evaluation across diverse datasets, making contributions easy to follow. (ceaX, NQQr)

We thank the reviewers for the reviews, and we will provide detailed responses to the specific weaknesses and questions in our official comments for each reviewer.

---

### Meta-Review · Area_Chair_1tR9 · 2024-12-15

**Metareview:**

This paper introduces a range-limited augmentation method for contrastive learning in tabular data and the FeSTa benchmark for few-shot learning on tabular datasets. The proposed method demonstrates improvements in preserving task-relevant information and achieving strong performance compared to existing methods.

However, after the rebuttal period, most reviewers still kept their negative ratings and tended to reject this paper. There are still important remaining concerns, e.g., the technical novelty of the proposed method is somewhat limited.

In the end, three reviewers give borderline and reject scores. Given the current weaknesses of the paper, the paper is rejected from the highly competitive ICLR conferences. The authors are encouraged to revise their manuscript according to the reviewers' comments and submit it to the next venue.

**Additional Comments On Reviewer Discussion:**

Reviewers' major concern is that the range-limited augmentation method lacks sufficient novelty, as it is an incremental improvement over existing methods. During the discussion period, Reviewer qfF3 confirmed their concerns were fully addressed. However, all other reviewers confirmed they still have remaining concerns.

---

### Decision · Program_Chairs · 2025-01-22

Reject